

# How bias correction goes wrong: Measurement of $X_{CO_2}$ affected by erroneous surface pressure estimates

Matthäus Kiel[1], Christopher W. O'Dell[4], Brendan Fisher[3], Annmarie Eldering[3], Ray Nassar[5], Cameron G. MacDonald[6], and Paul O. Wennberg[1,2]

[1]Division of Geological and Planetary Sciences, California Institute of Technology, Pasadena, CA, USA
[2]Division of Engineering and Applied Sciences, California Institute of Technology, Pasadena, CA, USA
[3]Jet Propulsion Laboratory, California Institute of Technology, Pasadena, CA, USA
[4]Colorado State University, Fort Collins, CO, USA
[5]Climate Research Division, Environment and Climate Change Canada, Toronto, ON, Canada
[6]Department of Physics and Astronomy, University of Waterloo, Waterloo, ON, Canada

**Correspondence:** Matthäus Kiel (mkiel@caltech.edu)

**Abstract.** Currently all measurements of $X_{CO_2}$ from space have systematic errors. To reduce a large fraction of these errors, a bias correction is applied to $X_{CO_2}$ retrieved from GOSAT and OCO-2 spectra using the ACOS retrieval algorithm. The bias correction uses, among other parameters, the surface pressure difference between the retrieval and the meteorological reanalysis. Erroneous surface pressure estimates, however, propagate nearly 1:1 into bias-corrected $X_{CO_2}$. For OCO-2, small errors in the knowledge of the pointing of the observatory (up to ∼130 arcsec) introduce a bias in $X_{CO_2}$ in regions with rough topography. Erroneous surface pressure estimates are also caused by a bug in ACOS version 8, sampling meteorological analyses at wrong times (up to three hours after the overpass time). Here, we derive new geolocations for OCO-2's eight footprints and show how using improved knowledge of surface pressure estimates in the bias correction reduces errors in OCO-2's v9 $X_{CO_2}$ data.

## 1 Introduction

Atmospheric carbon dioxide ($CO_2$) is currently being measured from space by, among other instruments, NASA's Orbiting Carbon Observatory 2 (OCO-2) and JAXA's Greenhouse gases Observing SATellite (GOSAT). Accurate and precise measurements of atmospheric $CO_2$ can identify and quantify its sources and sinks and, more generally, improve our understanding of biosphere-atmosphere fluxes. To do so, these measurements must be sufficiently accurate and precise to properly capture the small (<1%) spatial and temporal gradients of $CO_2$. OCO-2 and GOSAT $X_{CO_2}$ data have been widely used in studies to characterize fluxes from different sources, e.g. emissions from power plants (Nassar et al., 2017) or fires in Indonesia (Heymann et al., 2017). Other recent studies analyzed flux anomalies during El Niño periods (Liu et al., 2017).

OCO-2 and GOSAT share a common observational approach: solar reflectance spectra centered around 1.6 μm and 2.0 μm are used to determine the $CO_2$ optical depth. The $O_2$ optical depth is observed at the so-called 'A-band' centered around 0.7 μm. The column-averaged dry air mole fraction of $CO_2$ ($X_{CO_2}$) is determined by combining the information from these three spectral regions. The A-band is used to determine the amount of dry air along the $O_2$ optical path from the sun to the



spectrometer (airmass). The two $CO_2$ bands provide a measure of how many $CO_2$ molecules are in the (nearly) similar path. A fundamental challenge for the retrieval is that photons are scattered in the atmosphere and the efficiency of the scattering – primarily by clouds and aerosols – depends on wavelength. The wavelength dependent scattering is, in turn, estimated by the retrieval algorithm using information from both the $O_2$ spectra and the relative $CO_2$ optical depths determined from the two

different $CO_2$ bands.

Early analysis of $X_{CO_2}$ from the initial GOSAT spectra had global and regional systematic errors. Wunch et al. (2011b) demonstrated, however, that a large fraction of the error in $X_{CO_2}$ was correlated with retrieved components of the state vector in the retrieval algorithm. In particular, difference between the retrieval of surface pressure and that from the meteorological reanalysis was shown to correlate with error of similar magnitude in $X_{CO_2}$ (e.g. when the surface pressure retrieval was ∼1%

too large, the retrieved $X_{CO_2}$ was ∼1% too small). There are several reasons why surface pressure is not accurately retrieved from the spectra. First, errors in the knowledge of the spectroscopy of the oxygen can produce spurious airmass dependencies and can affect the pressure retrieval (e.g. Yang et al., 2005; Wunch et al., 2011a). Second, the algorithm is not adequately able to distinguish pathlength errors due to scattering from those due to surface pressure variation. For example, overestimates of the amount of aerosol near the surface (which shortens the path) can be compensated by an overestimate of surface pressure.

Because in the retrieval aerosols are generally assumed to scatter less efficiently at longer wavelengths, error in retrieved pathlength maps differently into $O_2$ and $CO_2$, resulting in a bias in $X_{CO_2}$.

Several retrieval codes that have been used to analyze GOSAT and OCO-2 spectra treat this problem differently. For example, the RemoTeC algorithm does not retrieve the surface pressure from the spectra. It uses the surface pressure from the meteorological reanalysis (Butz et al., 2011; Wu et al., 2018). Others, such as the University of Leicester Full Physics algorithm

(UoL-FP) first normalizes the retrieved $X_{CO_2}$ by the ratio of the retrieved surface pressure from the spectra and the surface pressure from the meteorological reanalysis. Then it uses the difference between the retrieved surface pressure and that from the meteorological reanalysis to 'bias correct' the $X_{CO_2}$ product (Cogan et al., 2012). To date, all versions of the Atmospheric Carbon Observations from Space (ACOS) retrieval algorithm (O'Dell et al., 2012; Crisp et al., 2012; O'Dell et al., 2018), used for both OCO-2 and GOSAT spectra, also have used the surface pressure difference between the retrieval and that diagnosed

from the meteorological reanalysis to bias correct the $X_{CO_2}$ product. This bias correction demonstrably improves the data set (Wunch et al., 2011b, 2017b; O'Dell et al., 2018). It also, however, places new demands on the accuracy of the meteorological analysis – demands that had not been considered at the time the OCO-2 mission was conceived. Error in the assumed pressure from the meteorological reanalyses at the field of view of the spectrometers will propagate nearly 1:1 into bias-corrected $X_{CO_2}$. Over land, for example, small errors in the knowledge of the pointing of the observatory can yield significant errors in esti-

mates of surface pressure in regions with rough topography. This is illustrated in Wunch et al. (2017b) where $X_{CO_2}$ variations near Lauder, New Zealand showed strong sensitivity to (different) estimates of the pointing of OCO-2, introducing an apparent topography related bias in the data. Finally, due to atmospheric tides, the estimate of the surface pressure is sensitive to when the meteorological reanalysis is sampled. Given the precision we are hoping to achieve in $X_{CO_2}$ measurements, seemly insignificant issues can not necessarily be ignored. For example, the mean canopy height of the Amazon rain forest is ∼25 m

(Benson et al., 2016) and might vary temporally due to fires or deforestation. Furthermore, the usual tidal range in the open





ocean is ∼0.5 m but coastal tidal ranges can reach up to 12 m (NOAA, last access: Aug. 2018). At sea level, altitude variations of ∼8 m correspond to changes in surface pressure of ∼1 hPa. This might introduce errors in $X_{CO_2}$ in the order of ∼0.4 ppm.

In this analysis, we address two issues with the OCO-2 v8 estimate of surface pressure: erroneous surface pressure values from the meteorological reanalysis due to small miss-specifications of the geolocations of OCO-2's eight footprints in

the instrument-to-spacecraft pointing offsets, and erroneous surface pressure estimates due to sampling the meteorological reanalysis at incorrect times. We illustrate how, using improved knowledge of the surface pressure, we can improve the bias correction and reduce errors in $X_{CO_2}$. The resulting hybrid product which uses v8 retrieval results with a revised bias correction using updated surface pressure estimates is labeled as version 9 (v9). This paper is structured as follows: section 2 describes the impact of erroneous surface pressure estimates in the bias correction on $X_{CO_2}$ estimates. New footprint geolocations for

OCO-2 are derived in section 3. Section 4 introduces the revised parametric bias correction in v9 and discusses changes in the v9 filtration scheme. Section 5 gives a brief evaluation of the OCO-2 v9 data product and illustrates changes and improvements of v9 over v8 $X_{CO_2}$ on regional and global scales.

## 2 Biases in OCO-2 $X_{CO_2}$ due to erroneous surface pressure estimates

OCO-2 v8 $X_{CO_2}$ estimates are derived using the ACOS retrieval algorithm. The algorithm uses optimal estimation to solve for

parameters of the state vector to obtain the best match to spectra recorded in OCO-2's three spectral bands. The state vector includes, among other parameters, the surface pressure which is primarily derived from information retrieved from the $O_2$ A-band. The prior surface pressure is taken from the GEOS-5 Forward Processing for Instrument Teams Atmospheric Data Assimilation System (GEOS5-FP-IT; Suarez et al., 2008; Lucchesi, 2013) and is sampled at the geolocation of each OCO-2 sounding. Surface pressure and prior surface pressure are used in the bias correction of $X_{CO_2}$. The OCO-2 bias correction

addresses three types of biases: footprint dependent biases, parameter dependent biases, and a global scaling of $X_{CO_2}$ to the World Meteorological Organization (WMO) trace-gas standard scale using comparisons to the Total Carbon Column Observing Network (TCCON; Wunch et al., 2011a). An overview of the three different bias correction terms is given in Mandrake et al. (2015), Wunch et al. (2017b), and O'Dell et al. (2018).

Biases in OCO-2 $X_{CO_2}$ due to erroneous surface pressure estimates were initially illustrated in OCO-2 observations over

Lauder, New Zealand (Fig. 10 in Wunch et al., 2017b). The Lauder TCCON site is situated in a remote area with no urban sources of $X_{CO_2}$ nearby (Pollard et al., 2017). The area is dominated by rolling hills with mountain ridges spanning from southwest to northeast, almost perpendicular to the ground-track of the observatory (southeast to northwest). The terrain changes up to ±200 m in altitude over small distances (see Fig. 1, upper panel). The middle panel of Fig. 1 shows $X_{CO_2}$ enhancements retrieved by the ACOS algorithm (version 8) over Lauder for a target observation on February 17, 2015. No bias correction

is applied here. $X_{CO_2}$ estimates are uniformly distributed over the observed scene with a mean value of 393.58 ppm and a standard deviation of 0.92 ppm. The lower panel of Fig. 1 shows OCO-2 $X_{CO_2}$ estimates after the v8 bias correction is applied. The bias correction changes the mean value to 395.95 ppm and increases the standard deviation to 1.35 ppm. Bias corrected $X_{CO_2}$ enhancements vary up to ±3 ppm over the observed scene. The bias is spatially correlated with the underlying topog-





raphy, more precisely, with the topographic slopes. The observed bias is introduced by erroneous values of the prior surface pressure in the dP term (the difference between the retrieved surface pressure and the prior surface pressure) in the parametric bias correction. The parametric bias correction accounts for spurious variability in $X_{CO_2}$ which correlates with retrieval parameters like albedo, retrieval aerosol quantities, or surface pressure. A multivariate regression is performed between spurious

$X_{CO_2}$ variability and the parameters that account for the largest variance in the data to correct for these errors (Wunch et al., 2011b; Mandrake et al., 2015; O'Dell et al., 2018). The erroneous values of the prior surface pressure are caused by small misspecifications in the geolocations of OCO-2's eight footprints in the specified instrument-to-spacecraft pointing. As stated previously, at sea level, a surface pressure difference of 1 hPa corresponds to an altitude difference of ∼8 m. Therefore, in areas like Lauder with steep topography, misspecifications in the pointing of the observatory of a few arcsec can cause the

prior surface pressure to be substantially different from the retrieved surface pressure. This introduces errors in bias-corrected $X_{CO_2}$, typically observed on local scales in areas with highly varying topography.

    Another source for erroneous surface pressure estimates in v8 is caused by a temporal sampling error of the surface pressure estimate from the meteorological reanalysis. The prior surface pressure is taken from the GEOS5-FP-IT three-hourly output. A bug in the meteorological sampling algorithm caused for some soundings the surface pressure estimate to be sampled as

much as three hours after the overpass time. This mostly affected soundings of orbits whose first and last sounding fully lies between synoptic GEOS5-FP-IT's three-hourly outputs (0z, 3z, etc); the soundings in such an orbit would be erroneously sampled at the upper bounding synoptic time for that orbit. For example, for an orbit whose soundings lie fully between 6:00 UTC and 9:00 UTC, the OCO-2 meteorological sampling algorithm erroneously samples the GEOS5-FP-IT surface pressure field at 9:00 UTC for each sounding in that orbit. On average, this introduced a mean prior surface pressure error of about +0.5

hPa for affected soundings. In some cases, however, the prior surface pressure error reached up to ± 20 hPa for individual soundings. The sampling error also affects temperature and water vapor. Soundings over land are affected more than over ocean since diurnal surface heating tends to be stronger over land and because the surface pressure bias correction term over land is nearly 50% larger than over water. While the sampling error of the prior surface pressure is easy to correct for via the bias correction by fixing the bug and re-running the meteorological sampling algorithm, erroneous surface pressure estimates

caused by misspecifications in the instrument pointing offsets need greater attention.

## 3    Evaluation of OCO-2's footprint geolocations

The core of the OCO-2 instrument is a three-channel grating spectrometer that records spectra of reflected sunlight in the $O_2$ A-band (0.7 µm), the weak $CO_2$ band (1.61 µm), and the strong $CO_2$ band (2.06 µm). The incoming light is guided through a common optics assembly but the light is sampled and focused sequentially and independently onto three spectrometer slits,

each 3 mm long and 28 µm wide (Haring et al., 2004; Crisp et al., 2017). These long, narrow slits are aligned to produce nominally co-boresighted fields of view. After passing the slit and being spectrally dispersed, the light is focused on a two dimensional focal plane array (FPA) with eight independent readouts along the slits - the so called footprints. Spectra for the three spectral bands and each footprint are recorded simultaneously.



To obtain the best estimate for the geolocation of the eight footprints, the following must be known: 1) the location of the spacecraft along the orbit track, 2) the pointing of the instrument boresight relative to a local coordinate system, and 3) the relative pointing of the fields of view (FOV) of the eight footprints in the three spectrometers. A Global Positioning System (GPS) sensor provides the location of the observatory along its orbit track. The on-board star tracker determines the orientation of the observatory relative to fixed stars. The relative alignment of the eight footprints is characterized with respect to the spacecraft body axes. The spatial FOV, defined along the long axis of the slit by the eight footprints, is aligned parallel with the spacecraft y-axis. The boresight of the spectrometer points down the -x-axis. The spacecraft z-axis points across the narrow axis of the spectrometer slit, perpendicular to the y-axis. For nadir and glint measurements, the z-axis is rotated around the -x-axis so it is oriented 30° (clockwise from above) from the principal plane (i.e. the plane that includes the sun, the surface target and the instrument aperture). To maintain this viewing geometry, the spacecraft slowly rotates counter clockwise (from above) around the -x-axis as it travels from the southern terminator, across the sub-solar latitude, to the northern terminator. South of a latitude that is $\sim 30°$ north of the sub-solar latitude, footprint 1 (FP 1) is to the west of footprint 8 (FP 8). North of this latitude, FP 1 is east of FP 8. For target mode observations, the z-axis is always pointed along the spacecraft orbit track, so that FP 1 is always to the west of FP 8. Pre-launch instrument ground-tests were performed to characterize the spatial FOV of each footprint and correction factors - the so called pointing offsets - have been derived and integrated into the geometric calibration algorithm (v0001 configuration, see Fig. 2). The pointing offsets are in the order of hundreds of arcsec. A change in the pointing offsets of, for example, 25 arcsec corresponds to a shift of the instrument FOV of ~80 m at nadir. During the OCO-2 in-orbit checkout (IOC) period in 2014, lunar measurements were performed and in combination with data from coastal crossings the alignment of the three spectrometer slits was tested. The alignment of the instrument angular footprints in the coordinate system defined by the star tracker was within mission requirements (< 720 arcsec). Updated pointing offsets have been integrated into the geometric calibration algorithm in November 2014 (v0006 configuration, see Fig. 2). The findings in the previous section, however, indicate that a reevaluation of the pointing vector correction factors is desirable.

## 3.1 Methodology

The analysis of the IOC lunar data exposed some deficiencies of its usage in elaborating footprint geolocations: lunar data is typically taken in so-called single pixel mode (when each pixel of the array is read out individually; this is in contrast to normal operations where 20 spatial pixel samples are co-added to form each footprint), the moon only illuminates a fraction of the FPA, defocus compromises the analysis of the strong $CO_2$ band results, and the moon only provides positive constraints for the z-axis. To overcome the aforementioned limitations for the v0006 configuration, the IOC lunar data results were used to constrain the pointing vector for FP 6 and 7, whereas for the other FPs the ground-test results were used. Here, we follow a different approach to derive new pointing offsets. We shift from estimating geolocations with lunar images, which are strictly geometric measurements, to optimizing footprint geolocations with retrieval variables. We utilize the ACOS Level 2 Full Physics (L2FP) algorithm and its associated pre-screeners, the A-band Preprocessor (ABP) and the IMAP-DOAS Preprocessor (IDP) to estimate footprint geolocations. The ABP performs a fast retrieval of surface pressure using the $O_2$ A-band and assumes that no clouds or aerosols are present. The IDP performs clear-sky fits to the weak and strong $CO_2$ bands to derive



$CO_2$ columns (Taylor et al., 2016). Using the preprocessors over the L2FP algorithm saves computational effort and allows us to study pointing offsets for each spectral band individually. The footprint geolocations for the $O_2$ A-band are derived by minimizing the variation in the difference between the surface pressure retrieved from the ABP and the meteorological analysis ($dP_{ABP}$). The location of the $CO_2$ band footprints is determined by minimizing the variation in the $CO_2$ columns divided by the dry air column determined from the meteorological analysis ($X_{CO_2,met}$). These two metrics are systematically explored for a set of different pointing offsets. The geolocations that provide the smallest standard deviation over a given scene for $dP_{ABP}$ are good estimates for the location of the $O_2$ A-band. The same holds for the standard deviation of $X_{CO_2,met}$ regarding the weak and strong $CO_2$ band. The assumption here is that there are no significant variations in $X_{CO_2}$ over the field of analysis. This may not be true in regions with large heterogeneous sources or sinks of $CO_2$ (e.g. urban areas). It is only true for areas with a clean $X_{CO_2}$ background. Therefore, in our analysis we focus on remote and mountainous areas to study pointing offsets.

## 3.2 Training Dataset

We identify two desert areas in the northern and southern hemisphere with topographic relief and frequent clear sky conditions during nadir and glint observations to derive new footprint geolocations: a remote area in the Death Valley National Park, CA, USA, and an area in the Atacama Desert, Chile. The Death Valley National Park area ranges from 35° to 37°N and from 118° to 115°W. The area in the Atacama Desert ranges from 18° to 19°S and from 69.8° to 69.25°W. Both areas are far from anthropogenic $CO_2$ sources. A topography related bias in v8 $X_{CO_2}$ is apparent in both areas (see Fig. 3). Observations over the Death Valley National Park include ∼1.8k soundings from September 2014 to September 2017. Observations over the Atacama Desert include ∼1k soundings from September 2014 to October 2017. All these soundings are aggregated into 0.02°×0.02° latitude-longitude grids. To account for the secular increase and seasonal cycle in $CO_2$ and different airmass values for different overpasses for each orbit, we normalize all $X_{CO_2,met}$ soundings by the orbital-mean. The standard deviation of $dP_{ABP}$ and $X_{CO_2,met}$ is calculated by taking into account all grid squares in the analyzed latitude and longitude limits. Analyzing data from both hemispheres allows us to check for possible errors introduced by the reversed orientation of the z- and y-axis in the northern and southern hemisphere in our pointing offset derivation (e.g. errors introduced by a timing error).

We run the ABP and IDP for a set of different pointing offsets for which the relative footprint positions of the v0006 configuration are preserved. If not otherwise stated, in the following we refer to the pointing offset of FP 4 of the $O_2$ A-band when we refer to pointing offset values. For example, if the pointing offset of FP 4 of the $O_2$ A-band is shifted by +25 arcsec along the y-axis, then all other footprint geolocations are also shifted in the same direction by +25 arcsec along the y-axis (even though their absolute positions differ from the FP 4 $O_2$ A-band position). The same holds for the z-axis. For the y-axis, we run both algorithms for four different pointing offsets ranging from 175 to 250 arcsec in 25 arcsec steps. For each of these shifts, we also run a set of different offsets for the z-axis, ranging from -250 to +100 arcsec also in 25 arcsec steps. This leads to a total of 60 different geolocation configurations.





### 3.3 Results

Figure 4 shows the standard deviation of $dP_{ABP}$ and $X_{CO_2,met}$ for FP 4 for all 60 geolocation configurations for the Death Valley National Park. The observed metrics are less sensitive to changes along the footprint axis than along the z-axis. Differences in the standard deviation between neighboring pointing offsets are small, typically $< 0.5$ hPa for the $O_2$ A-band and

$< 0.2$ ppm for the two $CO_2$ bands. This holds for all footprints in the three spectral bands. For example, for FP 2 to 7, the standard deviation of $dP_{ABP}$ is minimized for a pointing offset of 225 arcsec along the footprint axis. A pointing offset of 200 arcsec minimizes the standard deviation of FP 1 and 8. Similar results are derived for the Atacama Desert (not shown here). In general, a pointing offset of 225 arcsec along the footprint axis minimizes the standard deviation of $dP_{ABP}$ and $X_{CO_2,met}$ for the majority of the footprints. This offset value is nearly identical to the v0006 configuration (222.4 arcsec). Therefore, we

adapt a pointing offset of 225 arcsec along the y-axis for all footprints in the three spectral bands. The absolute pointing offsets along the footprint axis are summarized in Table 1.

Figure 5 shows the standard deviation of $dP_{ABP}$ and $X_{CO_2,met}$ as a function of the z-axis pointing offsets for FP 4 for the Death Valley National Park (for a pointing offset of 225 arcsec along the footprint axis). The analyzed metrics are strongly sensitive to changes of the pointing offset along this axis. We perform a quadratic regression to determine the best estimate of

the location of the minimum. We only take data points into account that are distributed symmetrically around the minimum. For FP 4, our analysis indicates a minimum at -124 arcsec for the $O_2$ A-band, -71 arcsec for the weak $CO_2$ band, and -44 arcsec for the strong $CO_2$ band. We derive pointing offsets for all other footprints for all three bands in the same way. Figure 6 (upper panel) summarizes the z-axis pointing offsets for all footprints for all three bands for the Death Valley National Park and Atacama Desert. On average, the derived pointing offsets for the two areas differ by 13 arcsec for the weak $CO_2$ band

and by 25 arcsec for the strong $CO_2$ band. For the $O_2$ A-band the differences between the two areas differ, on average, by 46 arcsec. Footprints 3 to 5 have the largest pointing offset values. This is in agreement with the relative footprint geolocations in the v0006 configuration. We average the derived pointing offsets for the $CO_2$ bands from both hemispheres. This provides the best estimate for the footprint geolocations globally and takes into account that the z-axis is rotated by nearly 180° (in glint and nadir mode) when the observatory overpasses the equator. However, for the $O_2$ A-Band, the difference between the

pointing offsets for both areas reaches up to 60 arcsec for FP 2. In addition, the Atacama Desert analysis indicate larger relative pointing variations for neighboring footprints. Therefore, for the $O_2$ A-band, we only take the derived pointing offsets from the Death Valley National Park analysis into account. Final pointing offsets for all three bands are derived by applying a quadratic regression to the pointing offsets as a function of footprint. This preserves the parabolic shape of the relative footprint positions which is supported by findings from the pre-launch and IOC lunar analysis. The updated pointing offsets for the z-axis for each

spectral band are summarized in Table 1.

To evaluate the impact of the updated footprint geolocations we sample the surface pressure from GEOS5-FP-IT with the updated meteorological sampling algorithm (that was corrected for the time sampling error) at the footprint geolocations of the $O_2$ A-band. The surface pressure is mainly retrieved from the $O_2$ A-band, therefore sampling the meteorological reanalysis at the $O_2$ footprint geolocation should yield best surface pressure estimates. Figure 7 shows the prior surface pressure difference



between v8 and sampled at the updated footprint geolocations. The striping pattern effect is mainly introduced by the updated sampling algorithm and follows orbital paths. As stated previously, the updated sampling method also introduces a mean bias of +0.5 hPa between the v8 and newly derived surface pressure estimates. Figure 8 shows the change between the standard deviation of the prior surface pressure in each grid box for both sampling methods. The observed structures are mainly driven

by changes in the footprint geolocations. The largest changes are over mountainous regions, e.g. the Tibetan Plateau, the Andes, or the U.S. West Coast. This will mostly manifest as local scale changes in $X_{CO_2}$. As expected, there are no significant changes over ocean due to the updated footprint geolocations.

## 4 The OCO-2 v9 data product

Our improved knowledge of OCO-2's footprint geolocations and the update of the meteorological sampling algorithm reduces

errors in bias-corrected $X_{CO_2}$ that were introduced through erroneous surface pressure estimates in the v8 bias correction. The OCO-2 v9 data product combines the v8 ACOS L2FP retrieval results with a revised bias correction using updated surface pressure estimates from GEOS5-FP-IT. Moreover, filter limits that define the $X_{CO_2}$ quality flag and warn levels are adjusted leading to a larger number of soundings that pass the filtration. Finally, the global scaling factor that is derived from direct observations over TCCON stations is updated. This section highlights the major changes in OCO-2's v9 $X_{CO_2}$. The techniques

that are used in the next sections are those presented in O'Dell et al. (2018). The derived results, with exception of the revised parametric bias correction, represent updates of the findings in O'Dell et al. (2018).

### 4.1 Parametric Biases Correction

The parametric bias correction accounts for spurious variability in $X_{CO_2}$ that is correlated with parameters in the retrieval state vector (Wunch et al., 2017b; O'Dell et al., 2018). A multivariate regression is performed between spurious $X_{CO_2}$ variations and

the parameters that account for the largest fraction of the spurious variability. For all ACOS versions for GOSAT and OCO-2 observations, the mode dependent parametric bias correction has the following form:

$$C_P = \sum_i c_i \left( p_i - p_{i,\mathrm{ref}} \right) \qquad (1)$$

Here, $c_i$ are regression coefficients, $p_i$ are the selected parameters, and $p_{i,\mathrm{ref}}$ the corresponding reference values. To select the parameters and derive the regression coefficients, different truth proxy training data sets were used for v8: TCCON, Small Area

Approximation (SAA), and Multi-Model Median. These truth proxies represent an independent estimate of $X_{CO_2}$ to which we compare OCO-2 $X_{CO_2}$. A detailed description of the truth proxies is given in Sect. 4.1 in O'Dell et al. (2018). For v8 land observations, three different parameters were identified that account for the largest fraction of variability: co2_grad_del, DWS, and dP. Over ocean, only co2_grad_del, and dP contribute to the parametric bias correction. co2_grad_del represents the tropospheric lapse rate of the retrieved $CO_2$ profile and is defined as the difference in the retrieved $CO_2$ between the surface

and the retrieval pressure level at 0.6 times the surface pressure, minus the same quantity for the prior profile. DWS represents the combined retrieved optical depth of large particles in the lower-to-middle troposphere in the retrieval, namely dust, water



cloud, and sea salt aerosol. In v8, $dP$ is defined as the difference between the retrieved surface pressure and the prior surface pressure from GEOS5-FP-IT.

For v9, we define two different $dP$ parameters for observations over land ($dP_{frac}$) and ocean ($dP_{sCO_2}$) that are used in the parametric bias correction. The revised $dP$ parameters take into account two problems: 1) the misspecifications in the geolocation calibration algorithm for the overall pointing of the observatory and 2) the pointing offsets between the three spectral bands. The first is characterized by the difference between the retrieved surface pressure of the v8 L2FP algorithm ($P_{ret,v8}$) and the prior surface pressure at the new geolocation where the $O_2$ A-band is pointing ($P_{ap,O_2}$). The second is characterized by the difference between the prior surface pressure where the $O_2$ A-band is pointing and the prior surface pressure where the strong $CO_2$ band is pointing ($P_{ap,sCO_2}$). For ocean, the revised $dP$ parameter has the following form (given in $hPa$):

$$dP_{sCO_2} = (P_{ret,v8} - P_{ap,O_2}) + (P_{ap,O_2} - P_{ap,CO_2})$$
$$= P_{ret,v8} - P_{ap,sCO_2} \qquad (2)$$

This approach allows us to reduce variations in $X_{CO_2}$ due to differences between the retrieved and estimated surface pressure without re-running the L2FP algorithm. Only the prior surface pressure sampled at the geolocation where the $CO_2$ bands are pointing is needed. Tests have shown that best results are achieved when the prior surface pressure is sampled at the geolocation of the strong $CO_2$ band. Over land, the revised $dP$ parameter accounts for the fractional change in $X_{CO_2}$ when error is present in surface pressure estimates (given in $ppm$):

$$dP_{frac} = X_{CO_2,raw} \left( 1 - \frac{P_{ap,sCO_2}}{P_{ret,v8}} \right) \qquad (3)$$

Here, $X_{CO_2,raw}$ represents the v8 $X_{CO_2}$ from the L2FP run when no bias correction is applied. A theoretical motivation for our choice of the $dP$ parameters over land and ocean is given in Appendix A. The definition of co2_grad_del and DWS remains the same in v9.

Similar to v8, we use three truth proxies to derive the parametric bias correction coefficients for co_grad_del, DWS and the revised $dP$ parameters (see Table 2). Compared to v8, the truth proxy data sets are extended in time to cover the longer OCO-2 data record. For the Multi-Model Median, nine models from the OCO-2 model-intercomparison project (MIP) are used (see Table 3). For all datasets a correction was applied using the OCO-2 averaging kernels based on Connor et al. (2008). We convolve the $CO_2$ profiles from the truth proxies with the OCO-2 column averaging kernel before we compare it to OCO-2 $X_{CO_2}$. The parametric bias correction coefficients for v9 are derived from the average of all coefficients derived from the different truth proxies. The adapted coefficients and reference values for land and ocean glint data are summarized in Table 4. The $dP_{frac}$ coefficient over land is close to 1. This is in agreement with the theoretical value since a change in surface pressure by ~1% changes $X_{CO_2}$ by also ~1% and seems to indicate that the retrieved surface pressure is still not sufficiently accurate to yield the best estimate of $X_{CO_2}$; indeed, as shown in $X_{CO_2}$, the coefficient implies that the optimal surface pressure is a weighted average of the retrieved and prior surface pressure, with the prior surface pressure weight being about 0.9. Figure 9 shows the different contributions of the v9 parametric bias correction to the raw $X_{CO_2}$.





## 4.2 Quality Filters

Bad soundings (e.g. those affected by clouds, low continuum level signal-to-noise ratio, etc.) are mostly screened out by the ABP and IDP before the ACOS L2FP algorithm performs retrievals. Some soundings that pass the pre-screening criteria, however, show errors in raw $X_{CO_2}$ when compared to the truth proxy training data sets that are too large to provide reliable

constraints on $CO_2$ fluxes. Therefore, threshold limits are defined for several variables to filter out these soundings. A detailed description on quality filtering is given Mandrake et al. (2015), Eldering et al. (2017), and O'Dell et al. (2018). We apply slight changes to the v9 filtration.

We introduce the new filter variables $dP_{O_2}$ and $dP_{sCO_2}$, the difference between the retrieved surface pressure and the estimated surface pressure at the geolocations of the $O_2$ A-band and $dP_{sCO_2}$ as given in Eq. (2). These variables replace the

dP filter variable in v8, which was defined as the difference between the retrieved surface pressure and a mean surface pressure estimate at the geolocation of all three spectral bands. The improved knowledge of the estimated surface pressure values allows us to relax the filter limits for the standard deviation of the surface elevation in the FOV. Figure 8 shows the bias and scatter in $X_{CO_2}$ over land relative to the Multi-Model Median truth proxy data set as a function of the standard deviation of the surface elevation. In v9, the scatter in the $X_{CO_2}$ difference starts to increase for standard deviations of the surface elevation larger

than 110 m whereas in v8 the scatter already increases for standard deviations larger than 60 m. Therefore, we extend the rather strict upper filter limit of 60 m in v8 to 110 m. This leads to a larger throughput of soundings in mountainous areas in v9. The parameters Max_Declocking_wco2 and Max_Declocking_sco2 are removed from the v9 filtration scheme over land. Moreover, filter limits for several other variables changed, e.g. rms_rel_wco2, $\tau_{oc}$, Band 3 albedo, and $dP_{ABP}$. The revised filter limits for rms_rel_wco2, $\tau_{oc}$, and Band 3 albedo cause a larger throughput for regions with boreal forests at high northern

latitudes. The updated limits for $\tau_{oc}$ and Band 3 albedo also increase the number of soundings over rain forests. The updated filter limits for $dP_{ABP}$ cause a larger throughput in regions with bright surfaces, e.g. the Saharan desert (see Fig. 11). Overall, 10-15% additional soundings pass the new filtration scheme compared to v8. All v9 filter variables and limits for land and ocean observations are summarized in Table 5. For soundings that pass filtration in both v8 and v9, the quality flag did not change.

## 25 4.3 Global Scaling factor

The global scaling factor corrects for an overall bias in $X_{CO_2}$ which still remains after filtration and application of the parametric bias correction. The global scaling factor is derived by comparing the OCO-2 data to TCCON measurements which are tied to the WMO scale (e.g. Wunch et al., 2010; Messerschmidt et al., 2010; Geibel et al., 2012). Due to changes in the data filtration and the revised parametric bias correction in v9, the global scaling factor $C_0$ needs to be updated, too. TCCON

stations that are used to derive the global scaling factor are listed in Table. 6.

We use the same geographic and temporal co-location criteria for OCO-2 data from direct overpasses of TCCON stations as in O'Dell et al. (2018). We apply the OCO-2 averaging kernels to TCCON data as discussed in the derivation of the coefficients in the parametric bias correction. The slope of the best fit line (forced through a zero intercept) is calculated using the method





described in York et al. (2004). The global scaling factor is roughly the same for the different observational modes over land and ocean. Ultimately, we adapt a value of 0.9954 over land and 0.9953 over ocean in v9 (compared with 0.9958 over land and 0.9955 over ocean in v8).

## 5   Brief evaluation of OCO-2 $X_{CO_2}$ data

Here, we evaluate the impact of the changes made in v9 on bias-corrected $X_{CO_2}$. To explore changes on local scales, we revisit the target observation over Lauder, New Zealand on February 17, 2015. Figure 12 shows both v8 and v9 bias-corrected $X_{CO_2}$. The improved knowledge of the prior surface pressure with the revised parametric bias correction clearly reduces the correlation between $X_{CO_2}$ and the underlying topography in v9. $X_{CO_2}$ values are distributed more uniformly over the observed scene. The standard deviation is reduced from 1.35 ppm in v8 to 0.74 ppm in v9. A small topography related bias is still apparent. However, compared to v8, it is a factor of two improvement in reducing biases caused by erroneous surface pressure estimates.

Figure 13 shows the absolute change in bias-corrected $X_{CO_2}$ between v8 and v9 globally. The observed changes are mainly driven by three factors: the updated meteorological sampling algorithm, improved knowledge of the footprint geolocations, and the revised parametric bias correction. In analogy to Fig. 7, the striping patterns follow orbital paths and are caused by the updated meteorological sampling algorithm. Differences over mountainous regions like the Tibetan Plateau or the Andes are driven by the improved knowledge of the prior surface pressure due to the updated footprint geolocations. The revised $dP_{frac}$ parameter in the parametric bias correction over land also introduces changes in regions at high altitudes but not necessarily with highly variable topography (e.g. South Africa). In addition, the v9 global scaling factor introduces a systematic difference of approximately +0.15 ppm between v8 and v9.

## 6   Conclusions

The update of the pointing vector that is used to derive the geolocation for OCO-2's eight footprints, together with an update of the meteorological sampling algorithm that corrects for a temporal sampling bug, provides a better estimate for the surface pressure in OCO-2's v9 data product. Biases in $X_{CO_2}$ due to erroneous surface pressure estimates are clearly reduced in regions with rough topography. For example, over Lauder, New Zealand, the standard deviation of bias-corrected $X_{CO_2}$ is reduced by almost a factor of two when the updated surface pressure estimates are used in the revised parametric bias correction that accounts for misspecifications in the instrument pointing offsets.

Accurate knowledge of the surface pressure and its estimate is crucial to retrieve $X_{CO_2}$ accurately and many challenges remain. The OCO-2 retrieval, for example, still has a latitudinally-dependent bias in surface pressure with a maximum in the tropics of nearly 5 hPa (O'Dell et al., 2018). Currently, it is thought that this originates in errors in describing the temperature dependence of the oxygen absorption. Moreover, uncertainties in the underlying elevation map and the question what the source





of the scattering is might have an impact on surface pressure estimates. This does not only affect $X_{CO_2}$ retrieved from GOSAT and OCO-2 but may also affect future sensors with similar observational approaches.

*Author contributions.* MK performed substantial data analysis regarding the derivation of new pointing offsets, revised bias correction, and global scaling factor for v9. CO was involved in nearly all aspects of this work, in particular with the revised bias correction, quality filtering, and global scaling factor for v9. BF implemented many tests and performed data analysis. AE provided project leadership and algorithm guidance. CM and RN helped to understand the origin of the topography related bias and contributed to the selection of the training datasets. PW provided critical guidance on nearly all aspects of the work, throughout all stages.

*Competing interests.* The authors declare that they have no conflict of interest.

*Acknowledgements.* We thank David Crisp for helpful discussions on the viewing geometry of the observatory. This work was financially supported by NASA's OCO-2 project (grant no. NNN12AA01C), and NASA's carbon cycle and ecosystems research program (grant no. NNX17AE15G).





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



**Table 1.** OCO-2 v9 instrument-to-spacecraft pointing offsets for each spectral band along the y-axis and z-axis relative to the central boresight of the telescope in units of arcsec.

|        |                 | FP1    | FP2    | FP3    | FP4    | FP5    | FP6    | FP7    | FP8     |
|--------|-----------------|--------|--------|--------|--------|--------|--------|--------|---------|
|        | O2 A-band       | 1339.5 | 972.4  | 597.2  | 225.0  | -140.6 | -508.1 | -879.6 | -1241.6 |
| y-axis | weak CO2 band   | 1331.0 | 966.2  | 604.7  | 235.9  | -131.5 | -499.4 | -860.8 | -1226.4 |
|        | strong CO2 band | 1359.1 | 987.3  | 614.7  | 244.0  | -125.9 | -496.0 | -867.1 | -1242.7 |
|        | O2 A-band       | -96.4  | -109.0 | -117.8 | -122.8 | -124.1 | -121.6 | -115.3 | -105.2  |
| z-axis | weak CO2 band   | -58.0  | -62.9  | -65.7  | -66.4  | -65.0  | -61.4  | -55.7  | -47.9   |
|        | strong CO2 band | -55.2  | -57.9  | -58.4  | -56.6  | -52.6  | -46.4  | -37.9  | -27.2   |



**Table 2.** Overview of the truth proxy training data sets for v9.

| Name | $N_s$ land | $N_s$ ocean glint | date | Details |
|---|---|---|---|---|
| TCCON | 614K | 360K | Sep. 2014 - May 2018 | GGG2014 (see Tab. 6) |
| Multi-Model Median | 956K | 2691K | Sep. 2014 - March 2017 | Median of 9 models |
| SAA | 63K | 287K | Sep. 2014 - Jan. 2017 | areas <100 km along-track |



**Table 3.** Models that contribute to the Multi-Model Median truth proxy data set.

| Name/Group | Version | Land/Biosphere | Inverse Method | Transport | Reference |
|---|---|---|---|---|---|
| CAMS | 15r2 | ORCHIDEE | 4D-Var | LMDZ | Chevallier et al. (2010) |
| CarbonTracker | CT2015,CT-NRT.v2016-1 | CASA | EnKF | TM5 | Peters et al. (2007) |
| TM5-4DVar-NOAA | 2016 | SiB-CASA | 4D-Var | TM5 | Basu et al. (2013) |
| OU | 2016 | CASA | 4D-Var | TM5 | Crowell et al. (2018) |
| Baker | | CASA-GFEDv3 | 4D-Var | PCTM | Baker et al. (2010) |
| CMS-Flux | | CASA-GFEDv3 | 4D-Var | GEOS-CHEM | Liu et al. (2017) |
| CSU-1 | | SiB4/MERRA | Bayesian Synthesis | GEOS-CHEM | |
| Jena CarboScope | s04_v3.8 | Special | 4D-Var | TM3 | Rödenbeck (2005) |
| Univ. Edinburgh | v2.1 | CASA | EnKF | GEOS-CHEM | Feng et al. (2009) |



**Table 4.** Parametric bias correction coefficients and reference values for v9 over land and ocean.

| land nadir/glint | $dP_{frac}$ | co2_grad_del | DWS |
|---|---|---|---|
| coefficients | -0.900 | -0.029 | -9.000 |
| reference values | 0.0 | 15.0 | 0.0 |

| ocean glint | $dP_{sCO_2}$ | max(co2_grad_del, -6) |
|---|---|---|
| coefficients | -0.245 | 0.090 |
| reference values | 0.0 | -6.0 |





**Table 5.** Filter variables and limits for the $X_{CO_2}$ quality flag definition in v9.

| variable | meaning | land filter | ocean filter |
|---|---|---|---|
| co2_ratio | Ratio of Band 2 to Band 3 $CO_2$ column from IDP algorithm | [1.00, 1.023] | [1.00, 1.02] |
| h2o_ratio | Ratio of Band 2 to Band 3 $H_2O$ column from IDP algorithm | [0.88, 1.01] | [0.88, 1.01] |
| $dP_{O_2}$ | Retrieved minus prior surface pressure [hPa] | [-8, 11] | [-5, 9] |
| $dP_{sCO_2}$ | Retrieved minus prior surface pressure [hPa] | [-10, 12] | [-5, 9] |
| $dP_{ABP}$ | Retrieved minus prior surface pressure from ABP algorithm [hPa] | [-12, 16] | [-50, 10] |
| windspeed | Retrieved surface wind speed [m/s] | | [1.5, 25] |
| co2_grad_del | Retrieved vertical gradient in $CO_2$ [ppm] | [-60, 85] | [-18, 30] |
| Altitude Stddev | Standard deviation of the surface elevation in the FOV [m] | [0, 110] | |
| Band 3 albedo | Retrieved albedo strong $CO_2$ band | [0.03, 0.6] | |
| albedo_slope_wco2 | Retrieved slope of the Lambertian component of the surface albedo using the strong $CO_2$ band $\left[cm^{-1}\right]$ | | [-1.5, 1.2]·1e-5 |
| albedo_slope_sco2 | Retrieved slope of the Lambertian component of the surface albedo using the strong $CO_2$ band $\left[cm^{-1}\right]$ | [-13, 100 ]·1e-5 | [0.6, 7]*1e-5 |
| rms_rel_wco2 | Relative RMS of Band 2 fit residuals [%] | [0, 0.28] | [0, 0.3] |
| rms_rel_sco2 | Relative RMS of Band 3 fit residuals [%] | [0, 0.45] | |
| $\tau_{total}$ | Retrieved optical depth of all aerosol types | [0, 0.5] | |
| $\tau_{WA}$ | Retrieved optical depth of water cloud | [0.0005, 0.1] | |
| $\tau_{IC}$ | Retrieved optical depth of ice cloud | [0.00, 0.04] | [0, 0.035] |
| $\tau_{ST}$ | Retrieved optical depth of stratospheric aerosol | [0.0002, 0.02] | |
| $\tau_{OC}$ | Retrieved optical depth of organic carbon | [0, 0.2] | |
| $\tau_{SS}$ | Retrieved optical depth of sea salt | [0, 0.125] | |
| $H_{IC}$ | Retrieved relative pressure height of ice cloud | [-0.5, 0.5] | |
| DWS | Retrieved optical depth of three large aerosol types (dust, water cloud, and sea salt) | [0, 0.25] | |
| eof33rel | Retrieved relative amplitude of third EOF of Band 3 | | [-0.3, 0.25] |
| $\chi^2_{wCO_2}$ | Reduced $\chi^2$ value of the L2FP fit residuals for Band 2 | | [0, 2] |
| $X_{CO_2,uncert.}$ | Posterior uncertainty in $X_{CO_2}$ [ppm] | | [0.28, 1.10] |
| Max_Declocking_wco2 | See O'Dell et al. (2018) for details | | [0, 0.27] |
| Max_Declocking_sco2 | See O'Dell et al. (2018) for details | | [0, 0.34] |





**Table 6.** Stations used in the TCCON truth proxy data set

| TCCON station | Reference | TCCON station | Reference |
|---|---|---|---|
| Anmyeondo, South Korea | Goo et al. (2014) | Lamont, OK, USA | Wennberg et al. (2016) |
| Ascension Island | Feist et al. (2014) | Lauder, New Zealand | Sherlock et al. (2014) |
| Bialystok, Poland | Deutscher et al. (2014) | Manaus, Brazil | Dubey et al. (2014) |
| Burgos, Phillipines | Velazco et al. (2017) | Ny Ålesund, Spitzbergen, Norway | Notholt et al. (2017) |
| Bremen, Germany | Notholt et al. (2014) | Orléans, France | Warneke et al. (2014) |
| Caltech, Pasadena, CA, USA | Wennberg et al. (2014b) | Paris, France | Te et al. (2014) (2014) |
| Darwin, Australia | Griffith et al. (2014a) | Park Falls, WI, USA | Wennberg et al. (2014a) |
| Edwards (Armstrong), CA, USA | Iraci et al. (2016) | Réunion Island | De Mazière et al. (2014) |
| East Trout Lake, Canada | Wunch et al. (2017a) | Rikubetsu, Japan | Morino et al. (2014b) |
| Eureka, Canada | Strong et al. (2017) | Saga, Japan | Kawakami et al. (2014) |
| Garmisch, Germany | Sussmann and Rettinger (2014) | Sodankylä, Finland | Kivi et al. (2014) |
| Izaña, Tenerife, Spain | Blumenstock et al. (2014) | Tsukuba, Japan | Morino et al. (2014a) |
| Karlsruhe, Germany | Hase et al. (2014) | Wollongong, Australia | Griffith et al. (2014b) |





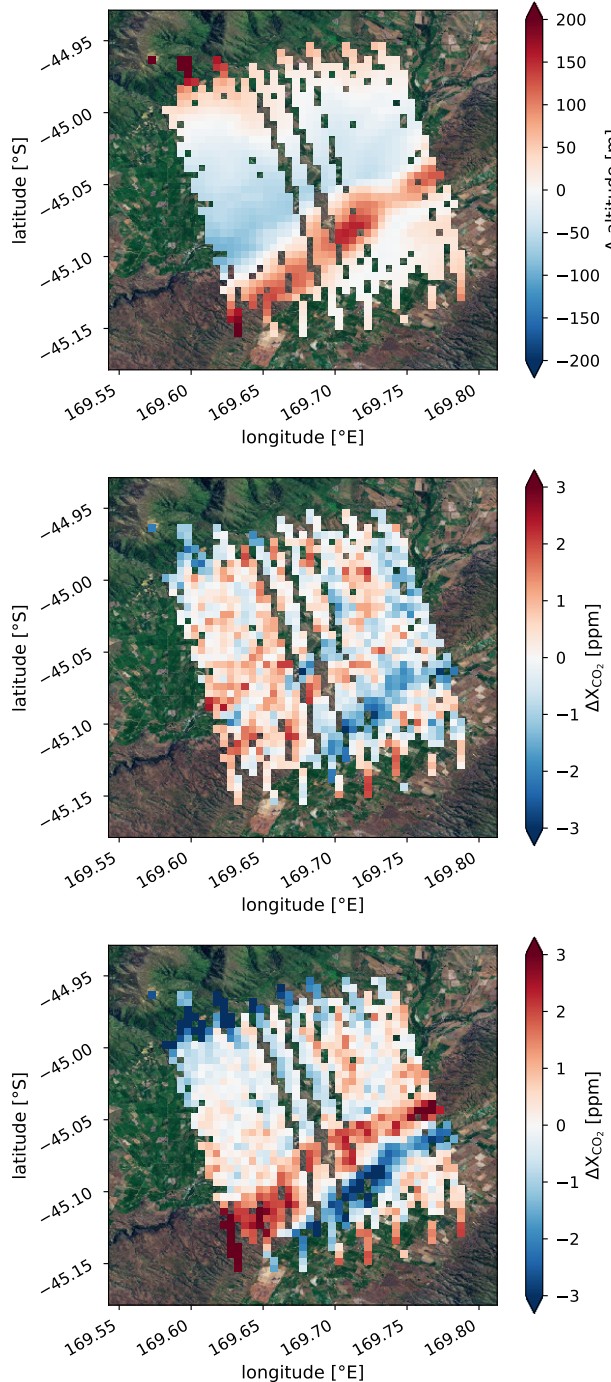

**Figure 1.** OCO-2 target mode observation over Lauder, New Zealand on February 17, 2015. The upper panel shows $\Delta$ altitude (defined as the sounding altitude minus the median altitude of all soundings in the given latitude and longitude limits). The middle and lower panel show the variation of raw and bias-corrected OCO-2 v8 $\Delta X_{CO_2}$ (defined in the same way as $\Delta$ altitude) after applying the v8 filters. Individual soundings are aggregated into $0.005° \times 0.005°$ latitude-longitude square grids.





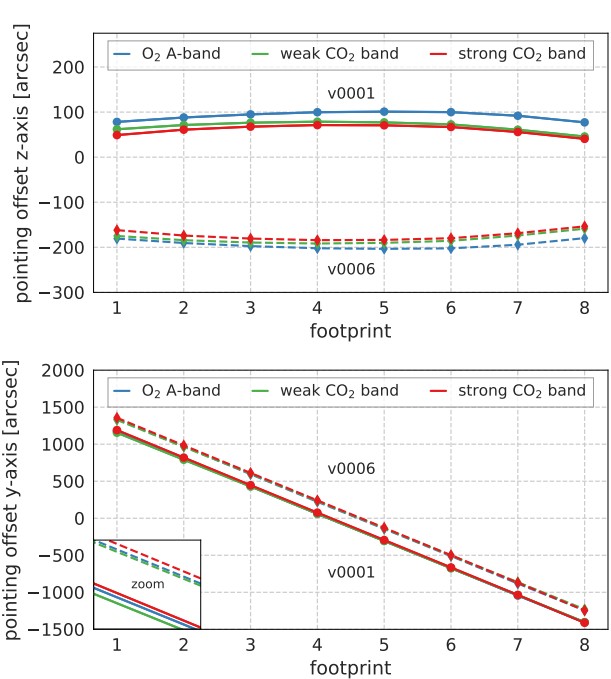

**Figure 2.** OCO-2 pointing offsets for each footprint and spectral band for the z-axis (upper panel) and y-axis (lower panel) derived from the pre-launch (v0001) and on-orbit (v0006) analyses.





**Figure 3.** Variation of altitude over the two selected areas in the Death Valley National Park (upper row) and Atacama Desert (lower row). Steep topography with altitude changes of up to $\pm 1000$ m are observed in both areas. A topography related bias in $\Delta X_{CO_2}$ derived from the ACOS L2FP retrieval is apparent in both regions. Individual observations are aggregated into $0.02° \times 0.02°$ latitude-longitude square grids.





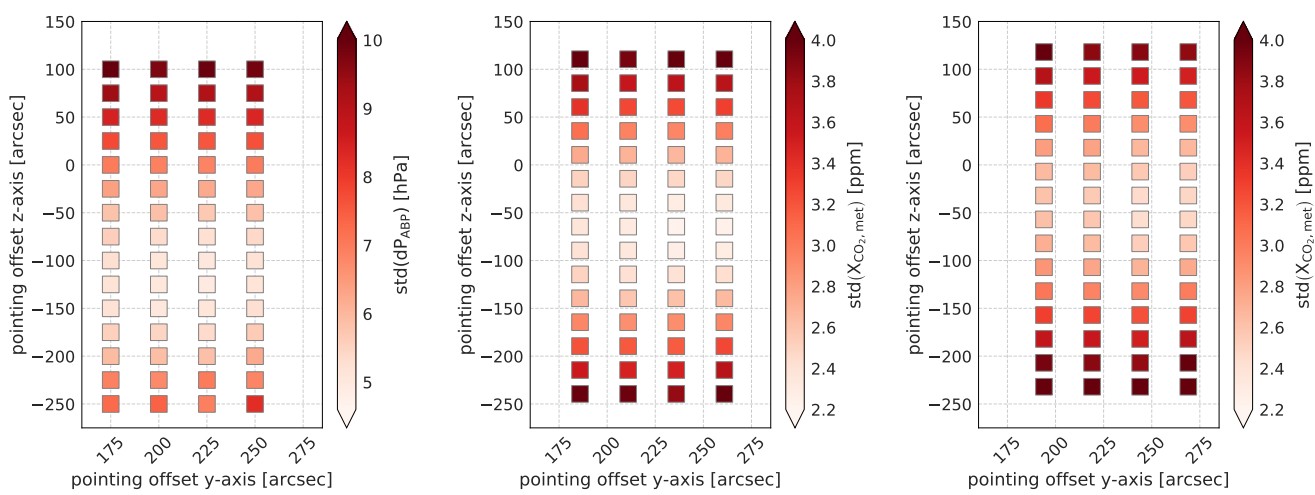

**Figure 4.** Standard deviation of $dP_{ABP}$ (left panel) and $X_{CO_2,met}$ for the weak (middle panel) and strong (right panel) $CO_2$ band for FP4 for all 60 geolocation configurations for the Death Valley National Park.





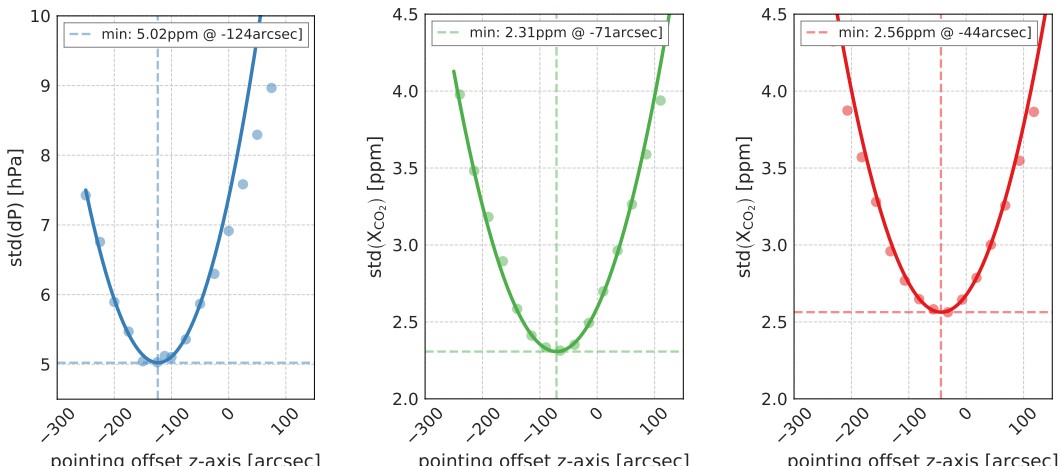

**Figure 5.** Standard deviation of $dP_{ABP}$ (left panel) and $X_{CO_2,met}$ of the weak (middle panel) and strong (right panel) $CO_2$ bands as a function of z-axis pointing offsets for FP4 for the Death Valley National Park. To determine the minimum, only values that are distributed symmetrically around the minimum are taken into account for the quadratic regression.





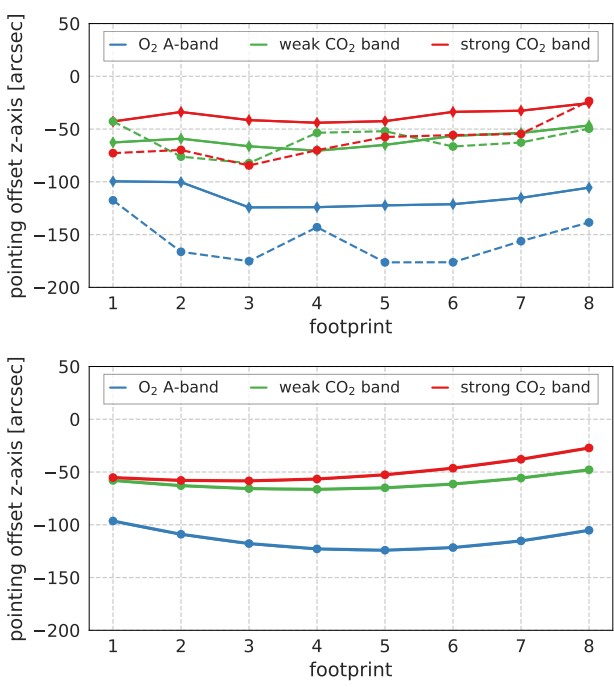

**Figure 6.** Upper panel: z-axis footprint pointing offsets for the three spectral bands for the Death Valley National Park (solid) and Atacama Desert (dashed); lower panel: z-axis footprint pointing offsets used in the OCO-2 v9 geometric calibration algorithm.





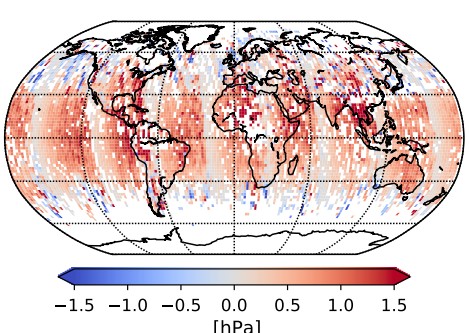

**Figure 7.** Mean difference between v9 and v8 (v9 - v8) surface pressure prior for April 2016. Data is aggregated into $2° \times 2°$ latitude-longitude square grids.



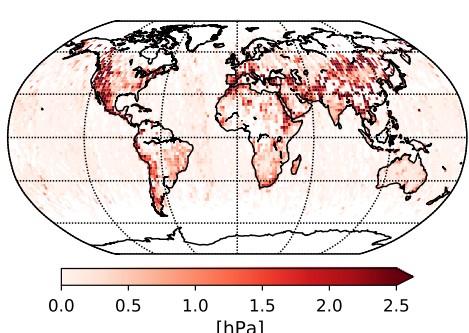

**Figure 8.** Difference between v9 and v8 (v9 - v8) of the surface pressure prior standard deviation in each grid cell for April 2016. Data is aggregated into $2° \times 2°$ latitude-longitude square grids.




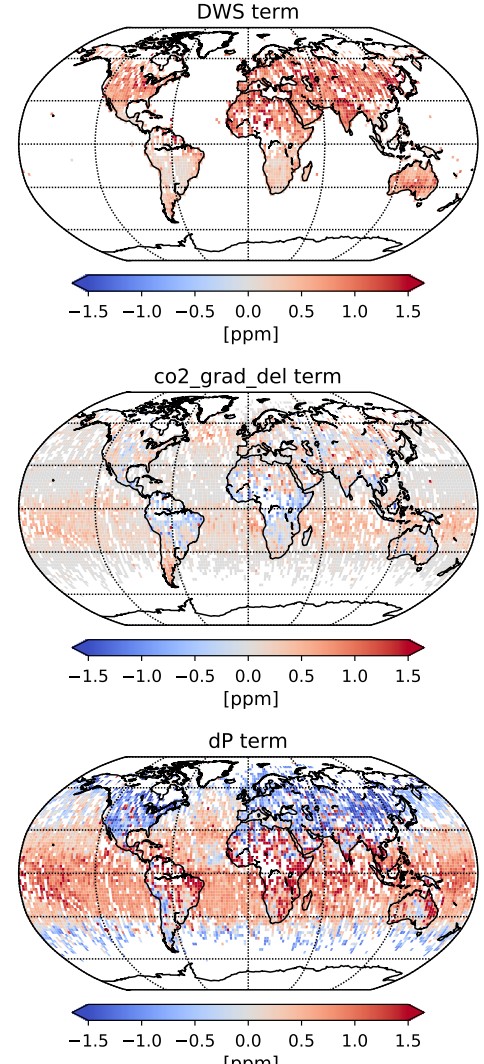

**Figure 9.** Contributions of the parametric bias correction terms to raw $X_{CO_2}$ from DWS (upper panel, only over land), co2_grad_del (middle panel), and the two dP terms over land and ocean (lower panel) for April 2016. Data is aggregated into $2° \times 2°$ latitude-longitude square grids.





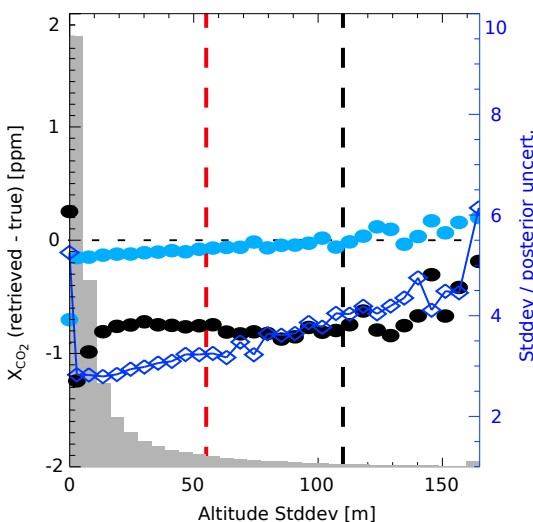

**Figure 10.** Difference between v9 $X_{CO_2}$ and the Multi-Model Median data set over land (nadir and glint) as a function of the standard deviation of the surface elevation in the FOV given in m. Shown are the mean bias, aggregated into 10 m bins, for both raw (black circles) and bias-corrected (light blue circles) $X_{CO_2}$. The standard deviation of the bias-corrected $X_{CO_2}$ difference is marked by dark blue diamonds. The distribution of the standard deviation of the surface elevation for the time period Sept. 2014 - March 2017 is shown in gray. The vertical black dashed line represents the v9 upper filter limit. The vertical red line represents the upper limit used in v8.



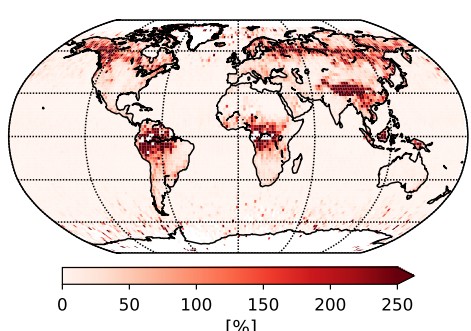

**Figure 11.** Relative increase of soundings that pass the v9 filtration scheme compared to v8 for the entire year 2016. Data is aggregated into $2° \times 2°$ latitude-longitude square grids.




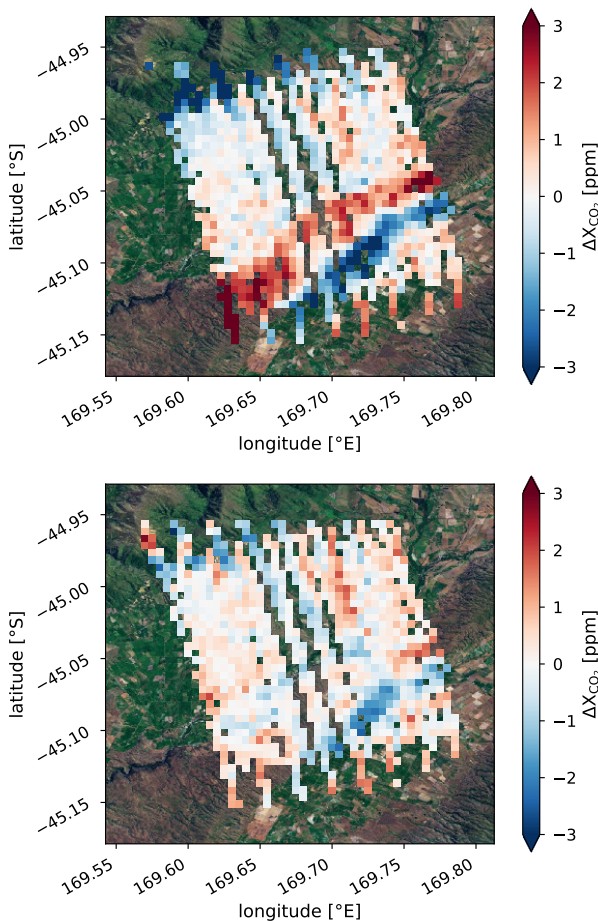

**Figure 12.** v8 (upper panel) and v9 (lower panel) bias corrected $X_{CO_2}$ over Lauder, New Zealand on February 17, 2015. $\Delta X_{CO_2}$ is defined in the same way as in Fig. 1. Data is aggregated into $0.005° \times 0.005°$ latitude-longitude square grids.





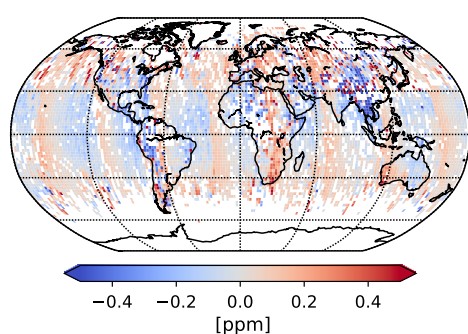

**Figure 13.** Global difference between v8 and v9 (v9 - v8) bias corrected $X_{CO_2}$ for April 2016. Only soundings that passed the v8 and v9 filtration are taken into account. A mean bias of 0.15 ppm (mainly introduced by the different global scaling factors for v8 and v9) is subtracted. Data is aggregated into $2° \times 2°$ latitude-longitude square grids.



**Appendix A: Theoretical motivation of dP parameters in the v9 parametric bias correction**

Column-averaged dry-air mole fractions of $CO_2$ are defined as the total column of $CO_2$ ($C_{CO_2}$) divided by the dry air column ($C_{dryair}$):

$$X_{CO_2} = \frac{C_{CO_2}}{C_{dryair}} \tag{A1}$$

$C_{dryair}$ is defined as:

$$C_{dryair} = \frac{P}{g_0 \cdot m_{dryair}} - \frac{C_{H_2O} \cdot m_{H_2O}}{m_{dryair}} \tag{A2}$$

Here, P is the surface pressure, $g_0$ the gravitational acceleration, $C_{H_2O}$ the total column of water vapour, $m_{dryair}$ the mean molecular weight of dry air, and $m_{H_2O}$ the mean molecular weight of water vapor. The surface pressure P can be written as:

$$P = c \cdot P_{ap} + (1-c) \cdot P_{ret} \tag{A3}$$

$P_{ap}$ and $P_{ret}$ represent the prior and retrieved surface pressure, respectively. The parameter $c$ is the fractional weight given to the prior in the assumed surface pressure. A value of $c = 0$ means that we completely trust the retrieval, $c = 1$ means that we completely trust the prior. For a start, we neglect the contribution of the total column of water vapor. Then the dry air column is directly proportional to the surface pressure and we can write:

$$X_{CO_2,raw} \propto \frac{C_{CO_2}}{P_{ret}} \tag{A4}$$

For bias-corrected $X_{CO_2}$ we can write:

$$X_{CO_2,bc} \propto \frac{C_{CO_2}}{c \cdot P_{ap} + (1-c) \cdot P_{ret}} = \frac{X_{CO_2,raw} \cdot P_{ret}}{c \cdot P_{ap} + (1-c) \cdot P_{ret}} = \frac{X_{CO_2,raw}}{c \cdot (P_{ap}/P_{ret}) + (1-c)} \tag{A5}$$

Taylor expansion leads to:

$$X_{CO_2,bc} = X_{CO_2,raw} + c \cdot \underbrace{X_{CO_2,raw} \cdot \left(1 - \frac{P_{ap}}{P_{ret}}\right)}_{dP_{frac}} \tag{A6}$$

The second term in Eq. (A6) is identical to the $dP_{frac}$ parameter that is used in the v9 parametric bias correction over land (see

Sect. 4.1). If we assume that relative variations in $X_{CO_2,raw}/P_{ret}$ are small compared to relative variations in $(P_{ret} - P_{ap})$, then we can further simplify to:

$$X_{CO_2,bc} = X_{CO_2,raw} + c \cdot (P_{ret} - P_{ap}) \tag{A7}$$

The second term of Eq. (A7) has the form of the $dP_{sCO_2}$ parameter as defined in Sect. 4.1. This form, however, does not account for the fractional change in $X_{CO_2}$ at higher elevations when error is present in surface pressure estimates. Therefore,

we use Eq. (A6) over land and Eq. (A7) only over ocean.