# Peer review of "How bias correction goes wrong: Measurement of $X_{CO_2}$ affected by erroneous surface pressure estimates"

_Atmospheric Measurement Techniques, 2018_

## Referee Comment (RC1) · Bréon (Referee) · 23 Nov 2018

Review of "How bias correction goes wrong: Measurement of XCO2 affected by erroneous surface pressure estimates" submitted for possible publication in AMT by M. Kiel et al.

This paper describes some recent refinements in the OCO-2 data processing that led to the generation of version 9 based on version 8 product. It is a must-read for those who use OCO-2 level 2 (XCO2) products and must understand the data processing and bias correction that has been applied to the data.

[Figure]

The paper is very clear, straightforward and well presented. It must be published with minor changes at the author discretion.

Minor suggestions : In the abstract (line 4), it is said that the pressure estimate error propagates 1:1 into XCO2. As XCO2 and Psurf do not share the same unit, it is not clear what that means. Adding the word "relative" may help (ie relative errors in Psurf translate nearly 1:1 into relative error in XCO2)

On a similar subject, in the introduction; line 10, a sentence could be added to state that the error transfer is somewhat expected as the CO2 measurement is sensitive to a number of molecule that is normalized by Psurf to deduce XCO2

At the end of section 3.1, one could add a sentence to state that, over vegetated areas, one could fear a different CO2 concentration in the low levels of the atmosphere with an impact on XCO2 over variable terrain. Thus, an analysis over desert area is preferable as is done in 3.2

P6 line 20. I could not understand the use of an "orbital mean'. What is that exactly, and why this choice rather than some information that is specific to the location and time of the observation?

---

## Referee Comment (RC2) · Anonymous Referee #2 · 16 Dec 2018

The paper by Kiel et al. reports on improved calculation of the a priori surface pressure used in the OCO-2 retrieval algorithm for the dry-air mole fraction of carbon dioxide (XCO2). Kiel et al. discovered two errors; one was related to erronous geolocation assignments, which in consequence caused wrong surface elevation going into the surface pressure calculation; the other error was related to wrongly interpolated pressure fields. The paper describes the errors and shows how to remedy them successfully in the next version of the algorithm. The paper should be published with minor modifications suggested below.

The study is probably of broader importance than just for OCO-2. GOSAT also suffered

[Figure]

from substantial uncertainties in its pointing information. The study clearly highlights the importance of accurate surface pressure knowledge – for all past and upcoming CO2 missions targeting at ppm accuracy. I would actually recommend highlighting a bit more that it is in particular localized analyses i.e. studies relying on pairs of geolocations (e.g. megacity urban dome vs. remote background) that require accurate geolocation assignment.

The paper repeatedly argues that it is the bias correction that causes wrong a priori surface pressure to map into wrong XCO2. So, the apparent remedy would be to trust in the retrieved surface pressure and not to bias-correct it (or not to use it for bias correction). The reason to retrieve surface pressure is actually based on the assumption that the a priori is not sufficiently accurate. Probably, that does not work because a) the retrieved surface pressure still must be heavily constraint to the a priori and b) retrieved surface pressure suffers from other errors (both due to the illposedness of the simultaneous aerosol retrieval, spectroscopic errors etc.). In summary, retrieving surface pressure does not lessen the need for accurate a priori information. That aspect could be made clearer in the manuscript.

P1,L6: "bug" -> (coding) error

P2,L12: It largely depends on surface albedo whether "too much aerosol" shortens or lengthens the lightpath. If surface albedo is high, multiple reflections between the surface and the aerosol layer are efficient and lengthen the path. I.e. the statement is not true in general.

P2,L13,14: Similar to the previous comment, spectral variation of surface albedo is probably even more important than spectral variation of aerosol optical properties in changing the radiative transfer regime between the O2A and the CO2 bands. Plus, the third player is the difference in absorption optical thickness structure between the bands that induces different height sensitivities to "wrong aerosol" when retrieving gas columns.

P2,L20: I recommend mentioning that, while ACOS has surface pressure in its state vector, it is heavily constraint to the a priori (I presume).

P2,L31: we are hoping to achieve -> we need to achieve

P4,L26: 0.7 -> 0.76

P5,1st paragraph: A sketch would help.

P6,L15+: 1.8k -> 1.800, 1k -> 1.000

Table 2: "K"? -> "/1000" in the header or at least "k"

Figure 3: To me, the topography related bias is not really apparent in the figures. Would it make sense to plot the slopes instead of the altitudes in the left panels?

App1,L17: Taylor expansion -> Taylor expansion in c around c=0 (right?)

App1, A7: So, strictly, the "c" in equ. (A7) is different from the "c" in equ. (A6).

---

## Author Comment (AC1) · 13 Jan 2019

**Author's reply to François-Marie Bréon**

We would like to thank François-Marie Bréon for helpful comments and suggestions. We will adapt all suggestions in the final version of the manuscript (amt-2018-353).

**Point-to-point response to specific comments and suggestions:**
* * *
**Referee:** In the abstract (line 4), it is said that the pressure estimate error propagates 1:1 into XCO2. As XCO2 and Psurf do not share the same unit, it is not clear what that means. Adding the word "relative" may help (ie relative errors in Psurf translate nearly 1:1 into relative error in XCO2).

**Authors:** We will rephrase the sentence in the following way:

*"Relative errors in the surface pressure estimates, however, propagate nearly 1:1 into relative error in bias-corrected $XCO_2$."*
* * *
**Referee:** On a similar subject, in the introduction; line 10, a sentence could be added to state that the error transfer is somewhat expected as the CO2 measurement is sensitive to a number of molecule that is normalized by Psurf to deduce XCO2.

**Authors:** We will add the following sentence to emphasize the normalization using $P_{surf}$:

*"$XCO_2$ is the ratio of $CO_2$ to the dry surface pressure. Any error that does not affect both, the $CO_2$ measurement and dry surface pressure, in the same way, is expected to propagate into $XCO_2$."*
* * *
**Referee:** At the end of section 3.1, one could add a sentence to state that, over vegetated areas, one could fear a different CO2 concentration in the low levels of the atmosphere with an impact on XCO2 over variable terrain. Thus, an analysis over desert area is preferable as is done in 3.2.

**Authors:** We modified the sentence in the following way:

*"The assumption here is that there are no significant variations in $XCO_2$ over the field of analysis. This may not be true in regions with large heterogeneous sources (e.g. urban areas) or sinks (vegetated areas) of $CO_2$."*
* * *
**Referee:** P6 line 20. I could not understand the use of an "orbital mean'. What is that exactly, and why this choice rather than some information that is specific to the location and time of the observation?

**Authors:** We added the following sentence:

*"The orbital mean is calculated by taking into account all soundings of a particular orbit that are within the latitude and longitude limits of the analyzed scene."*

---

## Author Comment (AC2) · 13 Jan 2019

**Author's reply to Referee #2**

We would like to thank the referee for helpful comments and suggestions. We will adapt all suggestions in the final version of the manuscript (amt-2018-353).

**Point-to-point response to specific comments and suggestions:**
* * *
**Referee:** P1,L6: "bug" -> (coding) error

**Authors:** We will replace *"bug"* by *"coding error"* throughout the manuscript.
* * *
**Referee:** P2,L12: It largely depends on surface albedo whether "too much aerosol" shortens or lengthens the lightpath. If surface albedo is high, multiple reflections between the surface and the aerosol layer are efficient and lengthen the path. I.e. the statement is not true in general.

**Authors:** We will add:

*"Pathlength errors also largely depend on surface albedo. For example, if the surface albedo is high, multiple reflections between the surface and the aerosol layer are efficient and lengthen the path."*
* * *
**Referee:** P2,L13,14: Similar to the previous comment, spectral variation of surface albedo is probably even more important than spectral variation of aerosol optical properties in changing the radiative transfer regime between the O2A and the CO2 bands. Plus, the third player is the difference in absorption optical thickness structure between the bands that induces different height sensitivities to "wrong aerosol" when retrieving gas columns.

**Authors:** We will add:

*"The spectral variation of surface albedo and aerosol optical properties also change the radiative transfer between the A-band and $CO_2$ bands. For example, differences in the absorption optical thickness structure between the three bands induce band dependent height sensitivities to different types of aerosols in the retrieval."*
* * *
**Referee:** P2,L20: I recommend mentioning that, while ACOS has surface pressure in its state vector, it is heavily constraint to the a priori (I presume).

**Authors:** We will mention that the surface pressure is substantially constrained by the surface pressure prior in the paragraph that discusses the elements of the ACOS state vector on P3, L15:

"*The state vector includes, among other parameters, the surface pressure which is primarily derived from information retrieved from the $O_2$ A-band (but substantially constrained by the surface pressure prior).*"
* * *
**Referee:** P2,L31: we are hoping to achieve -> we need to achieve

**Authors:** We will replace *"we are hoping to achieve"* by *"we need to achieve"*.
* * *
**Referee:** P4,L26: 0.7 -> 0.76

**Authors:** We will replace *"0.7"* by *"0.76"*.
* * *
**Referee:** P5,1st paragraph: A sketch would help.

**Authors:** We will add the following figure illustrating the orientation of the observatory:

[Figure]
* * *
**Referee:** P6,L15+: 1.8k -> 1.800, 1k -> 1.000

**Authors:** We will replace *"1.8k"* by *"1.800"* and *"1k"* by *"1.000"*.
* * *
**Referee:** Table 2: "K"? -> "/1000" in the header or at least "k"

**Authors:** We will add *"(x10³)"* in the header of Table 2.
* * *
**Referee:** Figure 3: To me, the topography related bias is not really apparent in the figures. Would it make sense to plot the slopes instead of the altitudes in the left panels?

**Authors:** We will add figures similar to the one below that show the slopes in the analyzed areas (e.g. in the figure below for the Death Valley scene Δaltitude represents the change in altitude in northeast direction which correlates with the bias in $XCO_2$).

[Figure]

$\Delta$ Altitude $\qquad$ $\Delta XCO_2$
* * *
**Referee:** App1,L17: Taylor expansion -> Taylor expansion in c around c=0 (right?)

**Authors:** We will replace *"Taylor expansion"* by *"Taylor expansion in c around c=0"*.
* * *
**Referee:** App1, A7: So, strictly, the "c" in equ. (A7) is different from the "c" in equ. (A6).

**Authors:** Yes, we will add the following sentences and refer to Table 4 where the different bias correction coefficients are listed:

*"Here, c represents the coefficient for the dP parameter in the parametric bias correction over land …. Note that the parametric bias correction coefficient c in Eq.(A6) and Eq.(A7) is different for land and ocean observations (see Tab. 4)."*

---

## Author Response (AR2)

**Author's reply to Christof Janssen**

We would like to thank Christof Janssen for helpful comments. We think that his suggestions will improve the quality of the manuscript. We adopted all of the suggestions in the final version of the manuscript (amt-2018-353).

**Point-to-point response to specific comments and suggestions:**
* * *
**Minor corrections:**

**Editor:** [0] Please check definitions, abbreviations, maths and units in your manuscript. The choice of variable names makes equation (1) seeming somewhat unrelated to equations (2) and (3). Is this so ? The naming of the variable dP_frac is unfortunate and confusing, because unlike dP_sCO2, dP_frac does not designate an absolute pressure difference but a relative pressure deviation multiplied by XCO2 (eq. (A6)). Please consider renaming it. The other confusion is on p. 39, where c once is a number defined in Eq. (A3), and then is used throughout equations (A4) to (A7), but in eq (A7), it suddenly denotes another variable with dimensions of 1/pressure. It would probably be easier to use another variable name (eg a) for the dimensionless constant in equations (A3 and subsequent) and then identify coefficient c in Eq. (1) with a in the case of Eq (A6) and with X_CO2,raw/P_ret*a in the case of eq. (A7). Moreover, it is not quite clear what the symbol C_P in Eq. (1) stands for (correction term with respect to parameter P ?). Wouldn't it be more straightforward to use X_CO2, or DeltaX_CO2 ?

**Authors:** We renamed the LHS of Eq. (1) to XCO2_para which represents the parametric bias. We also added a formula how to obtain bias corrected XCO2 by subtracting the parametric bias from raw XCO2 (see changes in manuscript below). The variable dP_frac is used in the official OCO-2 Data Product User's Guide for v9 (https://docserver.gesdisc.eosdis.nasa.gov/public/project/OCO/OCO2_DUG.V9.pdf), therefore we would like to keep the current notation. We changed 'c' to 'a' in the derivation of the dP parameters in Appendix A and by comparing Eq. (A6) with Eq.(2) in the revised manuscript we identify that c=-a.

**Editor:** [1] If we identify Eqs. (A6) and (1), and use the results from Table 4, it seems that c = -0.9. In this case the Taylor expansion around c=0 made to arrive at eq. (A6) is likely not valid.

**Authors:** Indeed, the Taylor expansion is performed in x=a(1-Pap/Pret) around x=0. We applied the necessary changes in Appendix A.

**Editor:** [2] Is it possible that the surface pressure is larger (or smaller) than both, P_ap and P_ret ? Then c should not be called fractional weight for which the constraint 0<=c<=1 would apply. Again, the fact that c = -0.9 is reported in Table 4 seems to indicate that this is the case, however.

**Authors:** See above and Appendix A in the revised manuscript.

**Editor:** [3] When going from Eq (A5) to (A6), you write "Taylor expansion in c around c=0

leads to:". Strictly speaking, this is not true, because (A5) is a proportionality, but eq (A6) an equality relation. Please correct.

**Authors:** Done.

**Editor:** [4] What is the significance of comparing rel. variations of X_CO2,raw/Pret with rel. variations in (P_ret-P_ap) (just before writing down Eq (A7))? Could you motivate any further?

**Authors:** The idea is that XCO2/Psurf has small variations over ocean, because over ocean XCO2 might vary from 390-410 ppm, and Psurf might vary from 995-1025 hPa. This variations are on the order of a few tenths of a percent on XCO2_para for the dP parameter, which itself is on the order of ~1 ppm. So tenths of a percent of 1 ppm is like ~0.003 ppm and therefore negligible over ocean.

**Editor:** [5] On p. 10 you write "The dP_frac coefficient over land is close to 1". In Table 4, you give a value of -0.9. Why don't you use that value in the text ? Intuitively, the minus sign does seem to be correct (an increased P reduces X_CO2). Does the departure from 1 come from the water in the atmosphere?

**Authors:** Done, we used a value of -0.9 in the text instead of "close to one". The dP_frac coefficient in the bias correction equation we use (where we are subtracting off the bias), has the opposite sign but same magnitude as "a".
* * *
*Technical corrections:*

**Editor:** p 1, abstract : consider to drop the word "currently" in "Currently all measurements of …"

**Authors:** Done.

**Editor:** p 1, abstract : "Relative errors in the surface pressure estimates, however, propagate nearly 1:1 into relative error in bias-corrected XCO2" -> "Relative errors in the surface pressure estimates, however, propagate nearly 1:1 into relative errors in bias-corrected XCO2"

**Authors:** Done.

**Editor:** p 1, abstract : "are also caused by an coding error" -> "are also caused by a coding error"

**Authors:** Done.

**Editor:** p 1, introduction : "The O2 optical depth is observed at the so-called ..." -> "The O2 optical depth is observed in the so-called …"

**Authors:** Done.

**Editor:** p 2, top. What is nearly similar and what would exactly similar mean ? Is it similar path or similar paths ? Please correct.

**Authors:** Due to an interband-offset in the pointing of the weak/strong $CO_2$ band and $O_2$ A-band (see Fig. 3), the optical paths from the sun to the spectrometer are only similar but not exactly the same for all three bands. We changed 'path' to 'paths'.

**Editor:** p 2, 2nd paragraph : "the spectroscopy of the oxygen" -> "the spectroscopy of oxygen"

**Authors:** Done

**Editor:** p 5, last paragraph before section 3.1: I guess there is no such thing as a -x-axis. Could you use positive or negative x-direction instead ? Similarly, "rotate about the x-axis" instead of "-x-axis"

**Authors:** Done.

**Editor:** p 5, section 3.1 : Please clarify first phrase which begins with "The analysis" and ends with "z-axis". It is very long and its logic is not very clear.

**Authors:** We simplified the sentence to: *"The analysis of the IOC lunar data exposed some deficiencies of its usage in elaborating footprint geolocations. Lunar data is typically taken in so-called single pixel mode when each pixel of the array is read out individually. This is in contrast to normal operations where 20 spatial pixel samples are co-added to form each footprint. In addition, the moon only illuminates a fraction of the FPA. Furthermore, defocus compromises the analysis of the strong $CO_2$ band results, and the moon only provides positive constraints for the z-axis."*

**Editor:** p 6, upper part : "Using the preprocessors over the L2FP algorithm". Does this mean "Using the preprocessors after the L2FP algorithm" ?

**Authors:** The preprocessor is run before the L2FP algorithm. Here we use only the preprocessor retrieval results instead of running the L2FP algorithm afterwards. The advantage of the preprocessors is that they provide single band retrievals which is desirable to derive the footprint geolocations for each individual band. We changed the sentence to: *"Using the preprocessors instead of the L2FP algorithm saves computational effort and allows us to study pointing offsets for each spectral band individually."*

**Editor:** p 6, "1.800 soundings" -> "1800 soundings"

**Authors:** Done.

**Editor:** p 6, "1.000 soundings" -> "1000 soundings"

**Authors:** Done.

**Editor:** p 8, 4.1 section title : Use singular as in text.

**Authors:** Done.

**Editor:** p 8, eq. (1) : index p on left hand side should be in small letters as it (probably) does not refer to pressure

**Authors:** We introduced the term XCO2_para instead. See 'Minor corrections' above.

**Editor:** p 8, eq. (1) : Introduce the symbol C_P at the left hand side in the text, and even better, replace it (see above).

**Authors:** Done. See 'Minor corrections' above.

**Editor:** p 8, eq. (1) c_i : please be more specific about the definition of the c_i. For instance, you could write c_i are regression coefficients which express the sensitivity of X_CO2 (C_P) to the selected parameter p_i.

**Authors:** We added the sentence: "*Here, a_i are regression coefficients which express the sensitivity of XCO2_bc to the selected parameter p_i, and p_i,ref are the corresponding reference values.*"

**Editor:** p 9, Third sentence after eq (3): "The definition of co2_grad_del and DWS remain the same in v9." -> "The definitions of co2_grad_del and of DWS remain the same in v9."

**Authors:** Done.

**Editor:** p 13, References : Please adopt the formatting recommendations for references and prefer the doi.org link over any other, eg:
Porter, J. G., De Bruyn, W., and Saltzman, E. S.: Eddy flux measurements of sulfur dioxide deposition to the sea surface, Atmos. Chem. Phys., 18, 15291–15305, https://doi.org/10.5194/acp-18-15291-2018, 2018. (attention, there are links beginning with "http:https:" (eg Cogan et al))

**Authors:** We removed URL entries (except for technical reports without DOIs) and refer to DOI entries where applicable.

**Editor:** p 22, Table 4 : units for coefficients and reference values are missing

**Authors:** We added units unless the coefficients/reference values are unitless.

**Editor:** p 23, Table 5, lines 3 and 4 : the meaning is the same even though variables are different. Please correct.

**Authors:** Done.

**Editor:** p 23, Table 5, line 10 : wco2 should correspond to the weak and not to the strong band

**Authors:** Done.

**Editor:** p 23, Table 5, line 11 : replace star by multiplication dot

**Authors:** Done.

**Editor:** p 24, Table 6 : Te et al. (2014)(2014) -> Te et al. (2014).

**Authors:** Done.

**Editor:** p 35, Figure 11 : please indicate in figure caption which data refers to the y-axis to the left and which to the scale on the right. The current choice of colors doesn't help.

**Authors:** We added the following sentence in the figure caption: *"Shown are the mean bias, aggregated into 10 m bins, for both raw (black circles) and bias-corrected (light blue circles) XCO2 (corresponding y-axis on the left). The standard deviation of the bias-corrected XCO2 difference is marked by dark blue diamonds (corresponding y-axis on the right)."*

**Editor:** p 39, right after eq (A2) : "mean molecular weight of water vapor" -> "molecular weight of water vapor"

**Authors:** Done.
* * *
Additional corrections by the authors:

p3, l23: In the first revised version we added that the surface pressure retrieval is substantially constrained by the surface pressure prior. This is not correct. The prior uncertainty on Psurf is 4 hPa (1-sigma) so it can pretty easily move within 10 hPa of the prior. We removed: "*(
[revised manuscript text omitted]

---

## Author Response (AR3)

**Author's reply to Christof Janssen**

**Point-to-point response to specific comments and suggestions:**
* * *
**Technical corrections:**

**Editor:** I point out that there is a misspelling in the reference to Rödenbeck (umlaut problem) and that the word "nearly" likely does not make any sense on top of p 2. It would be meaningful to speak about things that are nearly equal or very similar, but "nearly similar" sounds like very different. Do you intend to say that ?

**Authors:** We corrected the Rödenbeck reference. We removed the word 'nearly' similar on top of page 2.